# Ovariectomy uncouples lifespan from metabolic health and reveals a sex-hormone-dependent role of hepatic mTORC2 in aging

Sebastian I Arriola Apelo[1,2,3], Amy Lin[1,2,3], Jacqueline A Brinkman[2,3], Emma Meyer[1,2,3], Mark Morrison[2,3], Jay L Tomasiewicz[2], Cassidy P Pumper[2,3], Emma L Baar[2,3], Nicole E Richardson[2,3,4], Mohammed Alotaibi[2,3,4], Dudley W Lamming[2,3,4,5]*

[1]Department of Dairy Science, University of Wisconsin-Madison, Madison, United States; [2]William S. Middleton Memorial Veterans Hospital, Madison, United States; [3]Department of Medicine, University of Wisconsin-Madison, Madison, United States; [4]Endocrinology and Reproductive Physiology Graduate Training Program, University of Wisconsin-Madison, Madison, United States; [5]University of Wisconsin Carbone Comprehensive Cancer Center, University of Wisconsin, Madison, United States

**Abstract** Inhibition of mTOR (mechanistic Target Of Rapamycin) signaling by rapamycin promotes healthspan and longevity more strongly in females than males, perhaps because inhibition of hepatic mTORC2 (mTOR Complex 2) specifically reduces the lifespan of males. Here, we demonstrate using gonadectomy that the sex-specific impact of reduced hepatic mTORC2 is not reversed by depletion of sex hormones. Intriguingly, we find that ovariectomy uncouples lifespan from metabolic health, with ovariectomized females having improved survival despite paradoxically having increased adiposity and decreased control of blood glucose levels. Further, ovariectomy unexpectedly promotes midlife survival of female mice lacking hepatic mTORC2, significantly increasing the survival of those mice that do not develop cancer. In addition to identifying a sex hormone-dependent role for hepatic mTORC2 in female longevity, our results demonstrate that metabolic health is not inextricably linked to lifespan in mammals, and highlight the importance of evaluating healthspan in mammalian longevity studies.

*For correspondence:
dlamming@medicine.wisc.edu

## Introduction

The high frequency and co-morbidity of age-related disease in the aged limits the benefits that can be achieved by targeting them individually. An alternative approach, which could simultaneously treat or prevent many age-related diseases, is to target the aging process itself (*Kennedy et al., 2014*). However, extending lifespan without also extending healthspan is generally considered undesirable (*Kaeberlein, 2018*), and not all geroprotective interventions may necessarily improve healthspan. Indeed, a recent study in *C. elegans* found that lifespan and healthspan can be uncoupled in this organism (*Bansal et al., 2015*). Thus, there is a clear need to assess healthspan in mammalian longevity studies.

The mechanistic Target Of Rapamycin (mTOR) is a serine/threonine protein kinase that serves as a central regulator of metabolism and aging and is found in two distinct protein complexes, mTOR Complex 1 (mTORC1) and mTORC2 (*Kennedy and Lamming, 2016*). mTORC1 is acutely sensitive to the action of the FDA-approved pharmaceutical rapamycin, and inhibition of mTORC1, either genetically or by rapamycin, extends lifespan and healthspan in model organisms ranging from yeast

to mice (*Arriola Apelo et al., 2016b*; *Bitto et al., 2016*; *Harrison et al., 2009*; *Kaeberlein et al., 2005*; *Kapahi et al., 2004*; *Medvedik et al., 2007*; *Powers et al., 2006*; *Vellai et al., 2003*). Pharmaceutical or genetic inhibition of mTOR signaling tends to extend female lifespan to a greater extent than that of males (*Lamming, 2014*; *Miller et al., 2014*).

However, chronic treatment of mice with rapamycin induces a variety of metabolic side effects. We and others have shown that many of these effects are mediated by 'off-target' inhibition of mTORC2 (*Arriola Apelo et al., 2016a*; *Dumas and Lamming, 2020*; *Kleinert et al., 2017*; *Kleinert et al., 2014*; *Lamming et al., 2012*; *Sarbassov et al., 2006*; *Schreiber et al., 2019*; *Schreiber et al., 2015*). These studies highlighted an important gap in our knowledge – namely, the contribution of decreased mTORC2 signaling to the effect of rapamycin on lifespan. Studies in *C. elegans* and *D. melanogaster* have not convincingly addressed this question, as reduced mTORC2 signaling can be beneficial or detrimental in *C. elegans* depending on the tissue and diet, while over-expression of *Rictor*, which encodes an essential protein component of mTORC2, extends the lifespan of flies (*Chang et al., 2019*; *Mizunuma et al., 2014*; *Robida-Stubbs et al., 2012*; *Soukas et al., 2009*).

Deletion of *Rictor* in several distinct tissues or inducibly in the whole body of adult mice negatively impacts survival (*Chellappa et al., 2019*; *Lamming et al., 2014b*; *Yu et al., 2019*). Intriguingly, loss of hepatic mTORC2 specifically reduces male lifespan, with no effect of liver-specific deletion of *Rictor* on the lifespan of female mice (*Lamming et al., 2014b*). This difference between males and females was not due to sex-based differences in the efficiency of *Rictor* deletion (*Lamming et al., 2014b*). This sex-specific effect of hepatic mTORC2 inhibition may help to explain why rapamycin has a stronger effect on female lifespan (*Lamming, 2014*; *Miller et al., 2014*). Many sexually dimorphic responses, including to geroprotective interventions, are mediated in part by gonadal sex hormones (*Garratt et al., 2017*; *Garratt et al., 2018*). We hypothesized that gonadectomy would either protect male mice from the loss of hepatic *Rictor,* or sensitize female mice to the loss of hepatic *Rictor.*

We therefore examined how mice lacking hepatic *Rictor* responded to gonadectomy. We find that the male-specific negative impact of *Rictor* loss on overall survival persists following gonadectomy. While pre-pubertal gonadectomy negatively impacted the metabolic health of both liver *Rictor* knockout (L-RKO) mice and their wild-type littermates, it did not negatively impact survival, with ovariectomized mice in particular surviving longer than sham-surgery controls. Intriguingly, while as we expected based on our prior results, the overall survival of sham-surgery L-RKO female mice was not statistically different from that of female wild type controls, there was an apparent reduction in the survival of L-RKO Sham females between 400–800 days of age that was not observed in ovariectomized L-RKO mice. An in-depth analysis suggests that pre-pubertal ovariectomy promotes the midlife survival of female mice lacking hepatic *Rictor*, through protection from a non-cancer related cause of death. We conclude that there is a sex-specific role for hepatic mTORC2 in survival that is dependent upon sex hormones, and that ovariectomy uncouples metabolic health and longevity.

## Results

### Depletion of hepatic *Rictor* reduces the overall lifespan of male mice independently of sex hormones

We castrated male L-RKO mice and their wild-type littermates, and ovariectomized female L-RKO mice and their wild-type littermates, between 3 and 4 weeks of age; sham surgeries were performed on mice of both sexes and genotypes to control for any effects of the surgical procedures. We examined the longevity of all 8 groups of animals, separately analyzing females and males due to the expected sexually dimorphic effects. While cause of death analysis is difficult in mice, we performed gross necropsy on as many animals as possible, noting the presence or absence of observable cancers. As we expected for C57BL/6J mice (*Brayton et al., 2012*), we observed cancer in approximately half of the necropsied animals. There was no statistically significant difference in the frequency of observed cancer at necropsy between groups of male mice (chi-square test, p=0.06), although cancer was identified much more frequently in L-RKO CAST mice than in any other group of male mice (*Figure 1—figure supplement 1*).

We initially stratified our analysis of mice by the presence or absence of observable cancers at necropsy. We performed a Cox regression analysis to identify the overall effects of genotype and surgery, and then calculated the corrected two-sided log-rank sum p-value comparing individual curves for those effects identified as significant in the regression analysis. Intriguingly, there was no effect of hepatic *Rictor* deletion on the overall longevity of male mice that died with observed cancer (*Figure 1A*). In contrast, there was a dramatic effect of *Rictor* loss on the survival of male mice (HR = 4.84, p=0.0014) in which cancer was not observed during our gross necropsy (*Figure 1B*). We observed no overall effect of gonadectomy in either group of mice.

Looking at the survival of all of the male mice (*Figure 1C*), in agreement with our previous study (*Lamming et al., 2014b*) we observed an overall negative effect of hepatic *Rictor* loss on the survival of male mice, with deletion of *Rictor* leading to an increased hazard ratio (HR (RKO) = 1.75, p=0.01), and no overall effect of castration. This effect was driven in part by a statistically significant decrease (log-rank p=0.028) in the lifespan of intact male L-RKO mice relative to intact wild-type mice, with a 13.7% decrease in median lifespan. We also observed a 10% decrease in median lifespan in castrated male L-RKO relative to castrated male wild-type mice; these lifespan curves were not distinguishable (log-rank p=0.337). The almost complete overlap of the L-RKO Sham and L-RKO CAST survival curves demonstrate that castration does not rescue the lifespan defect of male mice lacking hepatic *Rictor*.

## Depletion of hepatic *Rictor* impairs midlife survival of female mice, a defect rescued by ovariectomy

There was no statistically significant difference in the frequency of observed cancer at necropsy between groups of female mice (chi-square test, p=0.78) (*Figure 2—figure supplement 1*), and we observed no effect of hepatic *Rictor* deletion on the overall longevity of female mice that died with observed cancer (*Figure 2A*). However, to our great surprise, there was a dramatic effect of *Rictor* loss on the survival of female mice (HR = 6.67, p=0.00084) in which cancer was not observed during our gross necropsy (*Figure 2B*). Although there was no overall effect of gonadectomy, we observed a strong interaction between the effect of *Rictor* loss and ovariectomy (p=0.014), and the survival curves of L-RKO OVX mice overlap the survival curves of WT Sham and WT OVX, leading us to interpret this interaction as demonstrating a strong protective effect of ovariectomy against a non-cancer related cause of death in mice lacking hepatic *Rictor*.

Despite this strong effect, when we looked at the total population of female mice in our study, the negative effect of *Rictor* loss on Sham-treated female mice was masked, and in agreement with our previous study (*Lamming et al., 2014b*) there was no overall negative effect of *Rictor* loss on survival (*Figure 2C*). There was also no difference between thesurvival of WT Sham and WT L-RKO female mice (p=0.153, Wilcoxon rank sum). We did however find an overall protective effect of prepubertal ovariectomy (HR (OVX) = 0.66, p=0.05). Inspection of the Kaplan-Meir plots and our results in *Figure 2B* lead us to conclude that this effect is largely driven by the protective effect of prepubertal ovariectomy on the midlife survival of female L-RKO mice. Finally, ovariectomized L-RKO female mice had a 17% increase in median lifespan relative to sham L-RKO females, due to a divergence of these lifespan curves between 400 and 800 days of age. This difference was statistically significant (p=0.025) by the Wilcoxon rank-sum test, which does not assume proportional hazards.

## Ovariectomy impairs the metabolic health of mice

During the course of our longevity study, we tracked weight and body composition (*Figure 3A–F*). Gonadectomy had a profound effect on the weight and body composition of wild-type mice, with castration reducing the weight of wild-type males due to a decrease in lean mass (*Figure 3A and C*), and with ovariectomy increasing the weight of wild-type females due an increase in fat mass (*Figure 3B and F*). Intriguingly, male L-RKO Sham mice weighed substantially less than their wild-type Sham littermates due to an overall decrease in fat mass (*Figure 3A and E*); however, no differences in fat mass due to genotype were observed in females (*Figure 3F*). Castration rescued the decreased adipose mass of L-RKO male mice, while not affecting fat mass in wild-type mice (*Figure 3E*).

In order to gain insight into the alterations in body weight, we used metabolic chambers to examine energy balance and fuel source utilization. We observed that ovariectomy, but not castration,

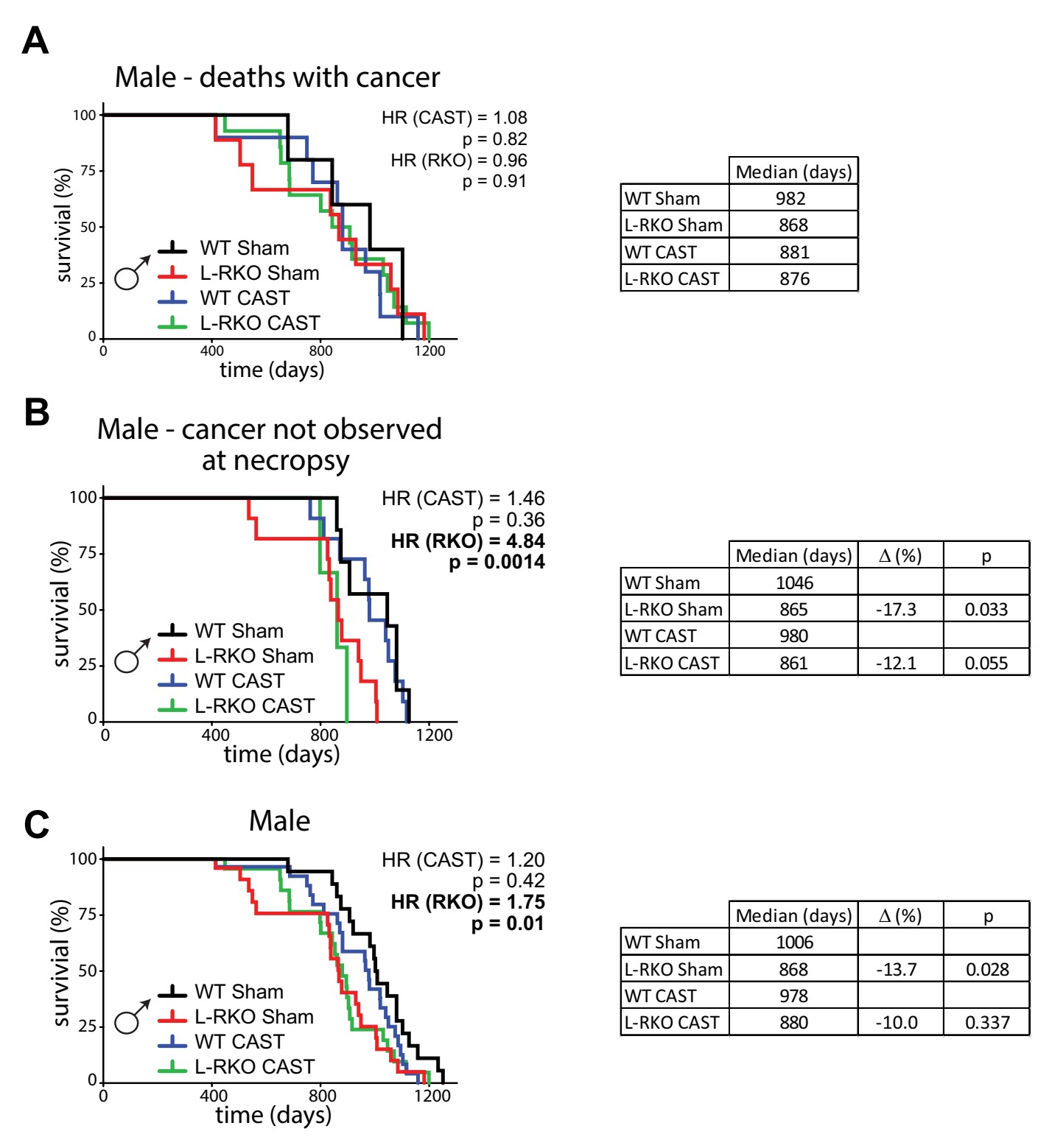

**Figure 1.** Deletion of hepatic *Rictor* impairs male survival independently of sex hormones. (**A**) Kaplan-Meier plot of the survival of male mice in which cancer was observed during gross necropsy (N = 38 male mice; WT Sham 5, L-RKO Sham 9, WT CAST 10, L-RKO CAST 14; **Supplementary file 1**). (**B**) Kaplan-Meier plot of the survival of male mice in which cancer was not observed during gross necropsy (N = 32 male mice; WT Sham 7, L-RKO Sham 11, WT CAST 11, L-RKO CAST 3; **Supplementary file 1**). (**C**) Kaplan-Meier plot of the survival of male mice lacking hepatic *Rictor* (L-RKO) and their wild-type (WT) littermates. All mice were subjected to gonadectomy (CAST) or Sham surgery at 3 weeks of age (N = 105 male mice; WT Sham 23,

*Figure 1 continued on next page*

*Figure 1 continued*

L-RKO Sham 27, WT CAST 29, L-RKO CAST 26; *Supplementary file 1*). (**A–C**) The overall effect of genotype (RKO), gonadectomy (CAST), and the interaction was determined using a Cox-proportional hazards test (HR, hazard ratio). The two-sided log-rank sum p-value was then calculated comparing individual curves for effects identified as significant in the regression analysis, and corrected for multiple comparisons (Holm-Sidak). The online version of this article includes the following figure supplement(s) for figure 1:

**Figure supplement 1.** Frequency of cancer observed at necropsy in male mice.

had profound effects on multiple components of energy balance and altered fuel source utilization (*Figure 4A–D*). In particular, ovariectomy suppressed food intake, activity, and energy expenditure in both WT and L-RKO mice (*Figure 4A,C,D*). Ovariectomy also altered fuel source utilization in WT females, with WT OVX mice having a lower RER indicating reduced oxidation of carbohydrates and increased utilization of lipids (*Figure 4B*). Interesting, L-RKO female mice had a lower RER during the light cycle that was not further suppressed by ovariectomy (*Figure 4B*).

We next examined how hepatic *Rictor* loss alters glucose tolerance and insulin sensitivity as the mice age in the presence and absence of gonadal hormones. As we expected based on our previous findings (*Lamming et al., 2014a*; *Lamming et al., 2014b*; *Lamming et al., 2012*), loss of hepatic *Rictor* had a profound negative effect on glucose tolerance in male mice (*Figure 5A and B*). Castration did not affect glucose tolerance in either WT or L-RKO mice. We observed no effect of hepatic *Rictor* deletion on the glucose tolerance of 4 month old female mice (*Figure 5C and D*), but 11 month old female L-RKO mice had impaired glucose tolerance relative to littermate controls. There was a clear overall negative effect of ovariectomy on glucose tolerance at both ages (*Figure 5C and D*).

We assessed insulin sensitivity by conducting insulin tolerance tests (ITTs, *Figure 5—figure supplement 1*) and by determining insulin resistance using homeostasis model assessment (*Levy et al., 1998*; *Figure 5E and F*). In males, consistent with impaired hepatic insulin sensitivity, deletion of hepatic *Rictor* resulted in fasting and glucose-stimulated hyperinsulinemia, and increased HOMA2-IR, despite negligible effects on the response to an ITT (*Figure 5E*, *Figure 5—figure supplements 1–2*); castration did not have a significant effect on any of these parameters. In contrast, in female mice genotype did not significantly affect fasting blood glucose or insulin, but ovariectomy did, promoting hyperinsulinemia and increasing HOMA2-IR (*Figure 5F*). We observed an effect of genotype on female ITT, with young L-RKO mice having a slight impairment of insulin sensitivity, and older ovariectomized L-RKO mice have a statistically significant impairment (*Figure 5—figure supplement 1*).

## Discussion

Here, we tested the role of sex hormones in the sexually dimorphic response to deletion of hepatic *Rictor,* which we previously observed to preferentially decrease the survival of male mice (*Lamming et al., 2014b*). We expected that if sex hormones were involved in this response, we would observe one of two possible outcomes: that castration would protect male mice from the loss of hepatic *Rictor*, or that alternatively, ovariectomy would sensitize female mice to the loss of hepatic *Rictor*. To our surprise, we observed neither effect; instead, castration failed to protect male L-RKO mice, while ovariectomy had a statistically significant beneficial effect on the midlife survival of female L-RKO mice, and may have had a mild positive effect on wild-type mice as well, increasing the median lifespan.

In contrast to our 2014 study, which found an effect of hepatic *Rictor* deletion only on male lifespan (*Lamming et al., 2014b*), here we also observed decreased midlife survival of female L-RKO mice. It is well appreciated that the lifespan of a mouse strain can vary substantially between laboratories (*Nadon et al., 2008*), and the relatively short lifespan of the wild-type control mice in our 2014 study, which was conducted in a different animal facility, may have obscured this difference. The reason for the decreased overall survival of male L-RKO mice and the decreased survival of female L-RKO mice during midlife is unknown; while our study suggests that this cause of death is not related to cancer, as we performed only a gross necropsy, we cannot at this time rule out the possibility that unobserved neoplasms contribute to these deaths. The mechanism by which ovariectomy protects L-RKO female mice during midlife is also unknown. Future carefully designed studies

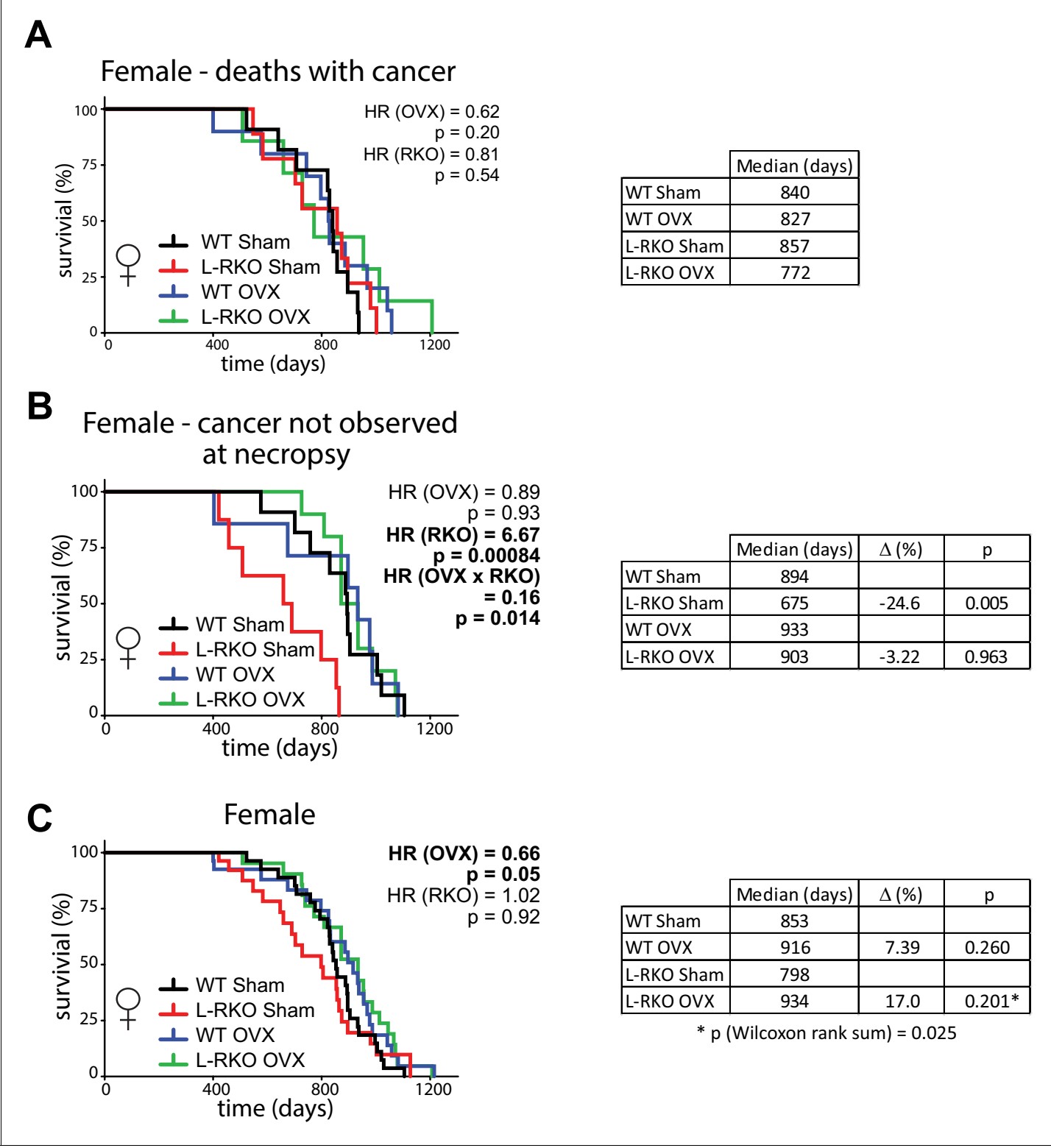

**Figure 2.** Ovariectomy protects female mice lacking hepatic *Rictor* during midlife. (**A**) Kaplan-Meier plot of the survival of female mice in which cancer was observed during gross necropsy (N = 37 female mice; WT Sham 11, L-RKO Sham 9, WT OVX 10, L-RKO OVX 7; *Supplementary file 1*). (**B**) Kaplan-Meier plot of the survival of female mice in which cancer was not observed during gross necropsy (N = 36 female mice; WT Sham 11, L-RKO Sham 8, WT OVX 7, L-RKO OVX 10; *Supplementary file 1*). (**C**) Kaplan-Meier plot of the survival of female mice lacking hepatic *Rictor* (L-RKO) and their wild-type (WT) littermates. All mice were subjected to gonadectomy (OVX) or Sham surgery at 3 weeks of age (N = 115 female mice; WT Sham 31, L-RKO

*Figure 2 continued on next page*

*Figure 2 continued*

Sham 29, WT OVX 27, L-RKO OVX 28; *Supplementary file 1*). (A–C) The overall effect of genotype (RKO), gonadectomy (OVX), and the interaction was determined using a Cox-proportional hazards test (HR, hazard ratio). The two-sided log-rank sum p-value was then calculated comparing individual curves for effects identified as significant in the regression analysis, and corrected for multiple comparisons (Holm-Sidak).

The online version of this article includes the following figure supplement(s) for figure 2:

**Figure supplement 1.** Frequency of cancer observed at necropsy in female mice.

to assess the organismal health of L-RKO mice, with detailed pathology between 400 and 800 days of age, will be needed to fully address these issues.

The role of sex hormones in the regulation of lifespan has been an active area of investigation for many years. Several studies have shown that early life ablation of the germline extends the lifespan of *C. elegans*, while ovariectomy extends the lifespan of *R. microptera* (*Arantes-Oliveira et al., 2002*; *Berman and Kenyon, 2006*; *Hatle et al., 2013*; *Iwasa et al., 2018*). Although some older studies suggest that ovariectomy decrease the lifespan of rats; at least one of these studies suffers from the lack of a sham surgery control group (*Asdell et al., 1967*); a more recent study found a small positive effect of ovariectomy on rat lifespan (*Iwasa et al., 2018*). A recent mouse study found that post-pubertal ovariectomy may shorten mouse lifespan (*Benedusi et al., 2015*); in contrast, mice in our study were ovariectomized prior to puberty.

Here, we find that pre-pubertal ovariectomy protects the midlife survival of L-RKO mice. Whereas Benedusi and colleagues report a reduction in female lifespan following ovariectomy, we found no statistically significant effect of pre-pubertal ovariectomy on the overall survival of wild-type mice, and in fact observed a small numerical increase (7.4%) in median lifespan. Our findings align with the results of model organism studies which suggest that early life ablation of the female germline promotes survival, and suggest that the control of aging by the female germline is conserved from nematodes and insects to mammals. Critically, the overall positive effects of ovariectomy on longevity do not promote metabolic health, with ovariectomized mice suffering from adiposity and disrupted blood glucose homeostasis, and showing reduced spontaneous activity and energy expenditure. The disruption of metabolic health we observed is in agreement many previous studies showing that ovarian hormones play a critical role in the control of glucose homeostasis in humans and mice (*Bailey and Ahmed-Sorour, 1980*; *Pirimoglu et al., 2011*; *Stubbins et al., 2012*; *Yuan et al., 2015*).

In contrast, we observed no effect of pre-pubertal castration on mouse lifespan. Our findings differ from results previously reported for rats and humans (*Drori and Folman, 1976*; *Min et al., 2012*), but these studies may have been impacted by effects of castration on rodent aggression or confounded by the social status of the castrated humans. Conversely, hormone replacement promotes the survival of men with late onset hypogonadism (*Comhaire, 2016*). Our results do not support a model in which ablation of the male germline is beneficial for healthspan or longevity. In agreement with recent studies by other groups (*Garratt et al., 2017*; *Harada et al., 2016*; *Inoue et al., 2010*), we observed an effect of castration on weight and body composition, but did not observe an effect of castration on glucose homeostasis.

A limitation of this study is that we did not perform a comprehensive assessment of healthspan (*Bellantuono et al., 2020*), focusing instead on metabolic parameters. Despite this, it is clear that as in the case of *C. elegans*, where lifespan and healthspan can be uncoupled (*Bansal et al., 2015*), pre-pubertal ovariectomy uncouples metabolic health and survival. Scientists around the world are now beginning to focus on geroprotective interventions as a way to prevent or treat age-related diseases. An open question surrounding such interventions has been the effect on health, as while there is a broad consensus in favor of extending healthspan, there is less support for extending lifespan without extending healthspan (*Kaeberlein, 2018*). Our research here highlights the need to comprehensively assess the effects of geroprotectors on healthspan, as at least some aspects of healthspan are not inextricably coupled to longevity.

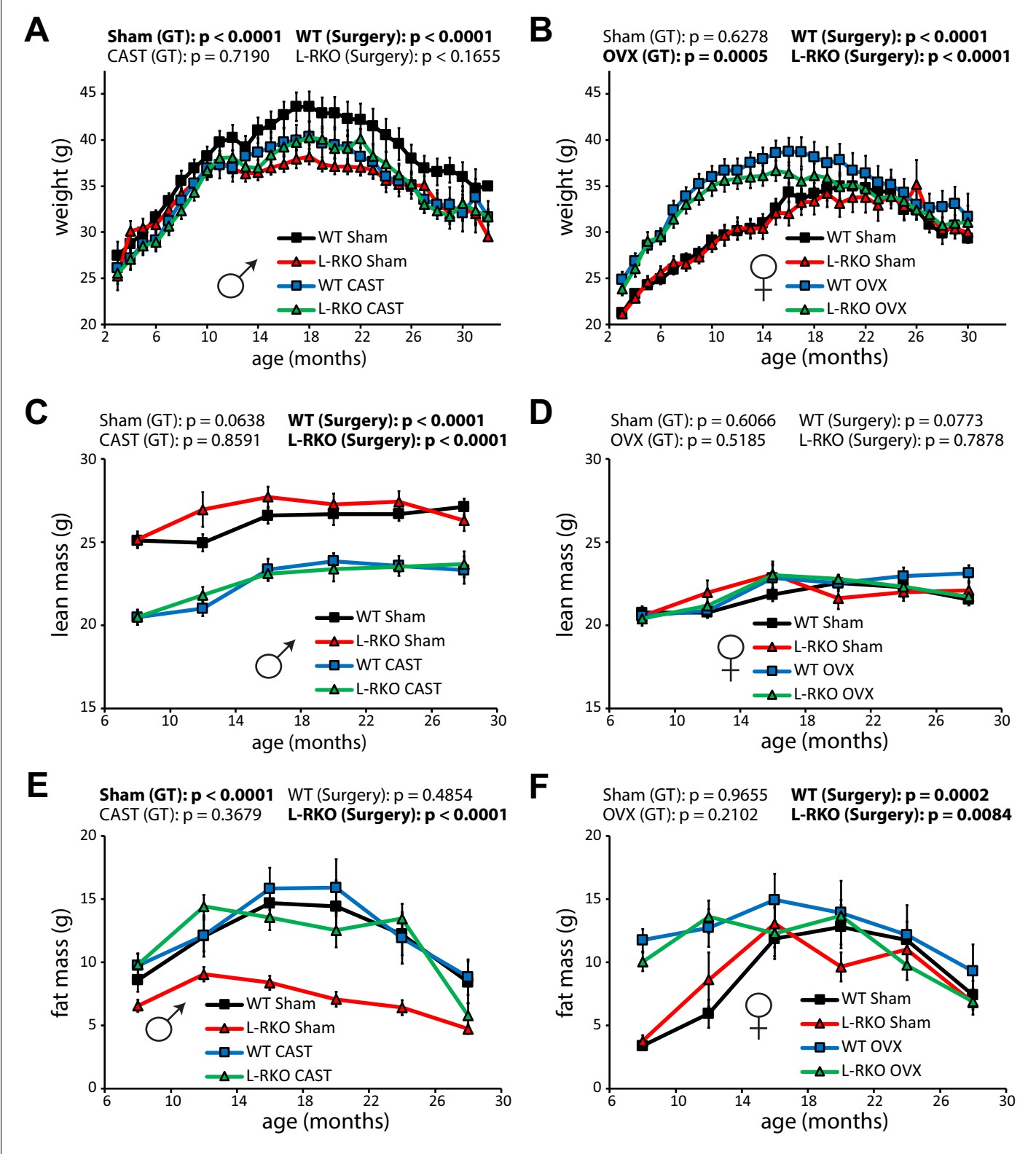

**Figure 3.** Gonadectomy affects weight and body composition, and rescues a male-specific effect of hepatic *Rictor* deletion on fat mass. (A, B) The weight of A) male and (B) female mice lacking hepatic *Rictor* (L-RKO) and their wild-type (WT) littermates was tracked starting at 3 months of age (n varies by month; maximum female N = 22–26 mice/group, maximum male N = 20–23 mice/group). (C–F) The lean mass (C, D) and fat mass (E, F) of mice was determined every 4 months starting at 8 months of age (n varies by month; max female N = 9–17/group; max male N = 12–17/group).

*Figure 3 continued on next page*

*Figure 3 continued*

p-values for the overall effect of genotype (GT) or surgical treatment (Surgery) represent the p-value from pairwise two-way ANOVA testing. Error bars represent SEM.

# Materials and methods

**Key resources table**

| Reagent type (species) or resource | Designation | Source or reference | Identifiers | Additional information |
|---|---|---|---|---|
| Genetic reagent (*Mus. musculus*) | C57BL/6J; *Rictor$^{loxP/loxP}$* | David Sabatini Lab (Whitehead Institute for Biomedical Research) | N/A | |
| Genetic reagent (*Mus. musculus*) | B6.Cg-*Speer6-ps1$^{Tg(Alb-cre)21Mgn}$*/J | The Jackson Laboratory | Stock number: 003574; RRID:IMSR_JAX:003574 | |
| Genetic reagent (*Mus. musculus*) | C57BL/6J; *Albumin-Cre; Rictor$^{loxP/loxP}$* | This paper | | See 'Animal use and care'; Dudley Lamming Lab (UW-Madison) |
| Drug | Human Insulin | Eli Lilly | NDC 0002-8215-17 (Humulin R U-100) | |
| Commercial assay or kit | Ultra-sensitive mouse insulin ELISA | Crystal Chem | Cat# 90080; RRID:AB_2783626 | |
| Software, algorithm | Prism 8 | GraphPad Software | N/A | |
| Software, algorithm | R (v. 3.6.0) survival package (v. 2.44) | *Therneau, 2015* | https://cran.r-project.org/web/packages/survival/index.html | |
| Software, algorithm | HOMA2 Calculator | *Levy et al., 1998* | https://www.dtu.ox.ac.uk/homacalculator/ | |
| Other | Normal Chow | Purina | Cat# 5001 | |

## Animal use and care

All animal procedures conducted at the William S. Middleton Memorial VA Hospital were approved by the Institutional Animal Care and Use Committee of the William S. Middleton Memorial Veterans Hospital (Assurance ID: D16-00403). Mice were multiply housed in microisolator cages and maintained under 12 hr light/dark cycles, and were fed Laboratory Rodent Diet 5001 (LabDiet). Mice hemizygous for *Albumin-Cre* and homozygous for a floxed allele of *Rictor* mice were obtained by crossing *Rictor$^{L/L}$* mice on a C57BL/6J background (*Lamming et al., 2012*), obtained from the Whitehead Institute for Biomedical Research (Cambridge, MA) and rederived by embryo transfer at the UW-Madison Biotechnology Center, with *Albumin-Cre* mice (The Jackson Laboratory, Stock 003574 *Postic et al., 1999*).

## Body composition

Body composition was measured by magnetic resonance imaging (EchoMRI, Echo Medical Systems, Houston, USA).

## Metabolic chambers

To assess food intake, RER, spontaneous activity, and energy expenditure by indirect calorimetry, we utilized an Oxymax/CLAMS metabolic chamber system following the manufacturer's instructions (Columbus Instruments).

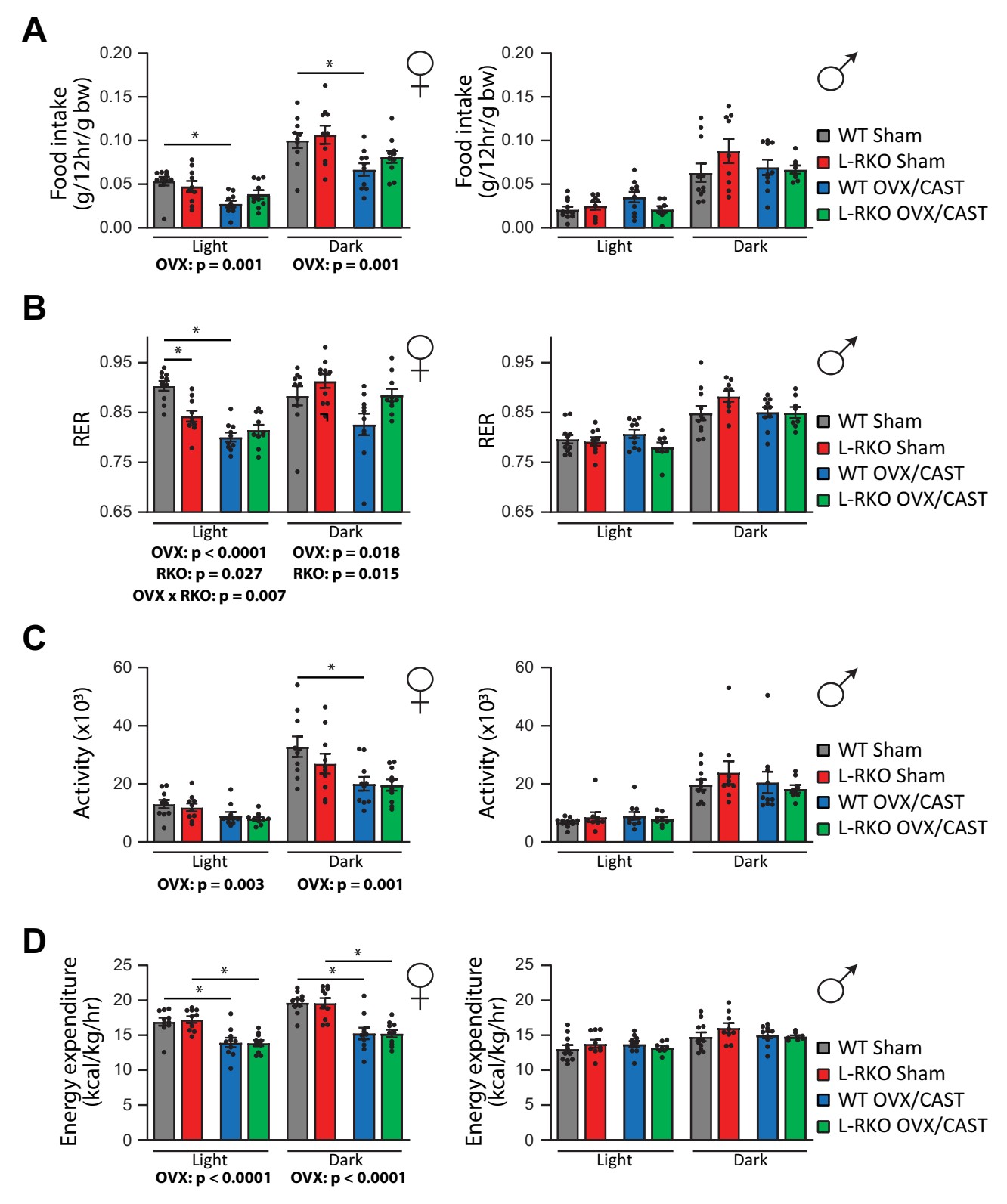

**Figure 4.** Gonadectomy alters energy balance and fuel source utilization. (**A–D**) Metabolic chambers were utilized to determine (**A**) food consumption, (**B**) Respiratory exchange ratio (RER), (**C**) Spontaneous activity, and (**D**) Energy expenditure in (left) female and (right) male mice lacking hepatic *Rictor* (L-RKO) and their wild-type (WT) littermates of the indicated surgical treatments at 12 months of age. The overall effect of genotype (RKO), gonadectomy (OVX or CAST), and the interaction represent the p-value from a two-way ANOVA conducted separately for the light and dark cycles; *=p < 0.05 from a

*Figure 4 continued on next page*

*Figure 4 continued*

Holm-Sidak post-test examining the effect of parameters identified as significant in the two-way ANOVA (n = Females, 10 mice/group; Males, WT Sham = 11 mice; WT CAST = 10 mice, L-RKO Sham = 9 mice, L-RKO CAST = 8 mice). Error bars represent SEM.

## Glucose and insulin tolerance tests, glucose-stimulated insulin secretion, and hormone measurements

Glucose tolerance tests were performed by fasting the mice overnight for 16 hr and then administering glucose (1 g/kg) intraperitoneally (*Arriola Apelo et al., 2016b*). Insulin tolerance tests were performed by fasting mice overnight for 16 hr, and then injecting 0.75 U/kg human insulin (Eli Lilly) intraperitoneally. Blood glucose was measured periodically for 2 hr after administration of glucose or insulin using a Bayer Contour blood glucose meter and test strips. For glucose-stimulated insulin secretion, mice were fasted overnight for 16 hr, blood glucose levels were determined and plasma was collected immediately prior to and 15 min after administering glucose (1 g/kg) intraperitoneally. Plasma insulin was quantified according to the manufacturer's protocol using an ultrasensitive mouse insulin ELISA kit (90080) from Crystal Chem. Fasting glucose and insulin levels were then used to calculate HOMA2-IR (*Levy et al., 1998*).

## Surgery

Mice were gonadectomized or subject to sham surgery during the third week of age. Following anesthesia with isoflurane, mice were shaved and the skin surface was sterilized with betadine and alcohol. For castration (or sham control surgery), a 1.5 cm ventral midline incision was made ending 0.5 cm cranial to the prepuce. In castrated animals, the vas deferens, located along the abdominal fat body on each side, was retrieved using a forceps or hemostat, and the testicle exteriorized. The spermatic artery was then clamped and ligated with 5–0 absorbable suture, and the testis removed. In both castrated and sham control mice, the body wall was closed with 5–0 monofilament absorbable suture, and staples used to close the skin incision. For ovariectomy (or sham control surgery), two 0.5 cm incisions were made through the skin, one on each side (1 cm ventral) to the paralumbar fossa area with its cranial terminus 1.5 cm caudal to the 13th rib, and any fascia trimmed away. Approximately 1 cm ventral to the dorsal spinous processes of the third lumbar vertebra, and immediately caudal to the last rib, the body wall was bluntly dissected through with a mosquito hemostat. For ovariectomy, each ovary was removed as follows; pressure on the abdomen was applied, causing the ovary to be extruded through one of the incisions (if the ovary did not exteriorize, a forceps was inserted to retrieve the ovary) just caudal to the kidney on that side. The ovary was then clamped for 30 s to the level of the fallopian tube, the ovary was removed, and the uterine horn was returned to the body cavity. In both ovariectomized and sham control mice, the muscles on both sides were then sutured with 5–0 absorbable suture, and the skin incisions closed with staples. For pain management, 0.1 mg/kg buprenex was administered IP to the mouse peri-operatively, and every 8–12 hr thereafter (at time of monitoring) as needed. Mice were allowed to recover on a heated pad for up to 1 hr following surgery, and then returned to the home cage. Post-surgery, female mice were provided with moistened breeder chow, and up to 1 mL per day of subcutaneous sterile saline if dehydrated. Staples were removed once healing was complete.

## Lifespan and necropsy

Following surgery, mice that successfully recovered were enrolled in the lifespan study at two months of age. A total of 115 female and 105 male mice were included in the lifespan study, with 23 female and 22 male mice removed for cross sectional analysis. Mice were co-housed (grouped by sex and surgical intervention) in specific pathogen-free housing. Mice were euthanized for humane reasons if moribund, if the mice developed other specified problems (e.g. excessive tumor burden), or upon the recommendation of the facility veterinarian. Mice found dead were noted at each daily inspection and saved in a refrigerator for gross necropsy, during which the abdominal and thoracic cavities were examined for the presence of solid tumors, metastases, splenomegaly, and infection; on the basis of this inspection the presence or absence of observable cancer was recorded.

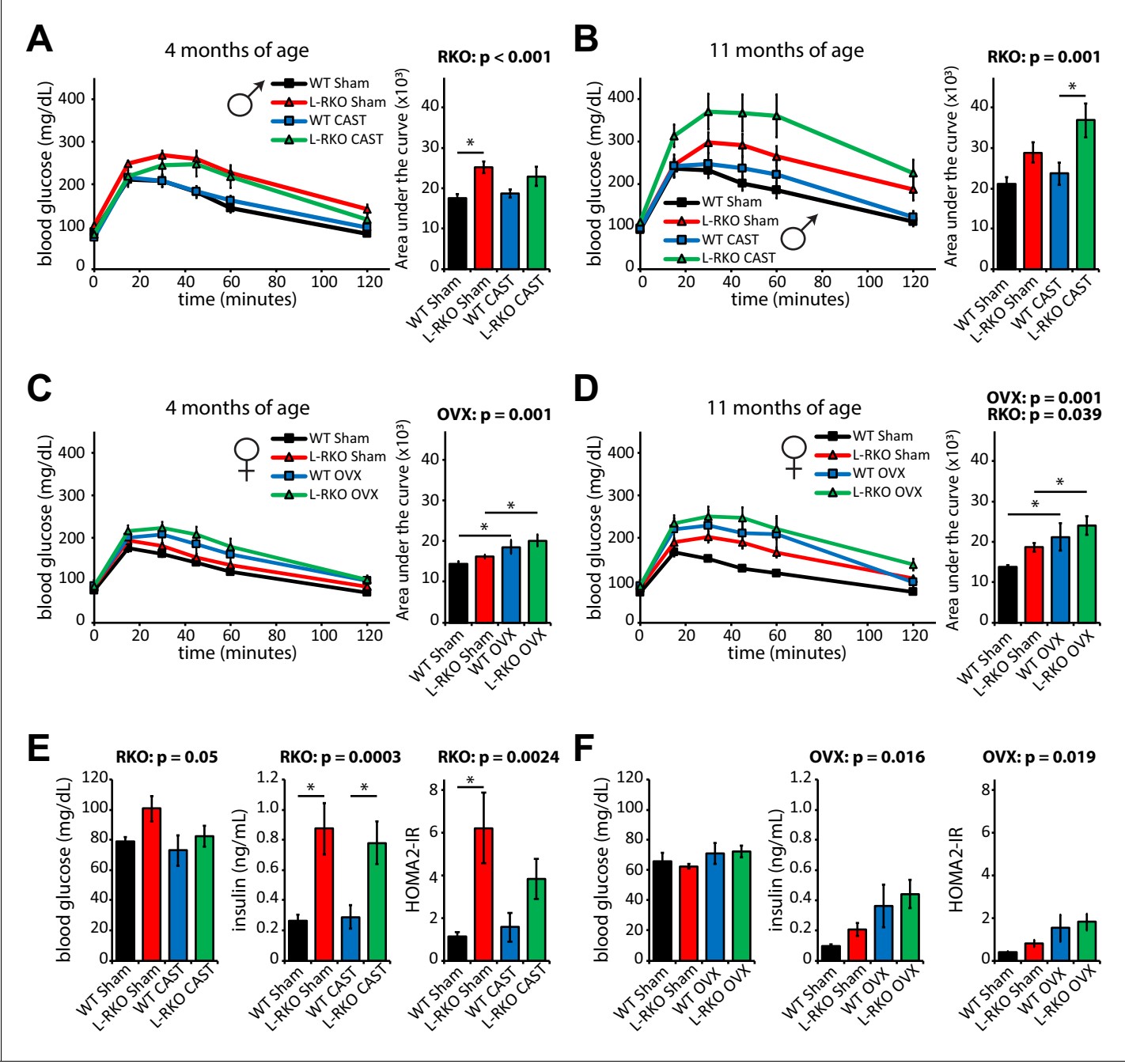

**Figure 5.** Independent effects of hepatic *Rictor* deletion and gonadectomy on glucose tolerance and insulin resistance. (A–D) The glucose tolerance of (A, B) male and (C, D) female mice lacking hepatic *Rictor* (L-RKO) and their wild-type (WT) littermates, of the indicated ages and surgical treatments, was determined following an overnight fast. Area under the curve: the overall effect of genotype (RKO), gonadectomy (OVX or CAST), and the interaction represent the p-value from a two-way ANOVA; *=p < 0.05 from a Holm-Sidak post-test examining the effect of parameters identified as significant in the two-way ANOVA (n = Males: A) 9–12 mice/group, 3–5 months of age; B) 7–12 mice/group, 10–12 months of age. Females: C) 7–11 mice/group, 3–5 months of age; D) 7–11 mice/group, 10–12 months of age). (E–F) fasting blood glucose and insulin values were used to calculate HOMA2-IR in E) male and F) female 14 month old mice (n = 4–5 mice/group; the overall effect of genotype (RKO), gonadectomy (OVX or CAST), and the interaction represent the p-value from a two-way ANOVA; *=p < 0.05 from a Holm-Sidak post-test examining the effect of parameters identified as significant in the two-way ANOVA). Error bars represent SEM.

The online version of this article includes the following figure supplement(s) for figure 5:

**Figure supplement 1.** Effect of hepatic *Rictor* loss and gonadectomy on insulin tolerance.

**Figure supplement 2.** Effect of hepatic *Rictor* loss and gonadectomy on glucose stimulated insulin secretion.

## Statistical analysis

Data are expressed as mean ± s.e.m. Statistical analysis was conducted using Prism 8 (GraphPad Software), except for survival analyses were conducted in R (version 3.5.0) using the 'survival' package (version 2.38) (T, 2015). Cox proportional hazards analysis was performed separately for each sex using genotype and surgical treatment as covariates. Mice which were removed from the study for cross-sectional analysis, and mice for which no record of age of death was available, were censored as of the last day for which a record was available.

## Acknowledgements

We thank Dr. Abigail Radcliff for advice on animal surgery and the University of Wisconsin-Madison Biotechnology Center Transgenic Animal Facility for rederivation services. We thank A Broman of the UW-Madison Department of Biostatistics and Medical Informatics for assistance with analysis of lifespan data and R. We thank Dr. Joseph Baur and Dr. Maria Mihaylova for critical reading of the manuscript, and Dr. Dena Cohen for support and encouragement. This research was supported in part by an American Federation for Aging Research Junior Faculty Research Grant to DWL. Additional research support was provided by the National Institute of Health/National Institute on Aging (AG041765, AG050135, AG051974, AG056771, AG062328), a Glenn Foundation Award for Research in the Biological Mechanisms of Aging to DWL and funds from the University of Wisconsin-Madison School of Medicine and Public Health and Department of Medicine to DWL. SIAA was supported in part by a fellowship from the American Diabetes Association (1–16-PMF-001). NER was supported in part by a training grant from the UW Institute on Aging (NIA T32 AG000213). The project was supported by the Clinical and Translational Science Award (CTSA) program, through the NIH National Center for Advancing Translational Sciences (NCATS), grant UL1TR002373. The Lamming laboratory is supported in part by the U.S. Department of Veterans Affairs (I01-BX004031), and this work was supported using facilities and resources from the William S Middleton Memorial Veterans Hospital. The content is solely the responsibility of the authors and does not necessarily represent the official views of the NIH. This work does not represent the views of the Department of Veterans Affairs or the United States Government.

## Additional information

### Competing interests

Dudley W Lamming: DWL has received funding from, and is a scientific advisory board member of, Aeovian Pharmaceuticals, which seeks to develop novel, selective mTOR inhibitors for the treatment of various diseases. The other authors declare that no competing interests exist.

### Funding

| Funder | Grant reference number | Author |
| --- | --- | --- |
| American Federation for Aging Research | | Dudley W Lamming |
| National Institute on Aging | AG041765 | Nicole E Richardson<br>Dudley W Lamming |
| American Diabetes Association | 1-16-PMF-001 | Sebastian I Arriola Apelo |
| National Center for Advancing Translational Sciences | UL1TR002373 | Dudley W Lamming |
| Glenn Foundation for Medical Research | | Dudley W Lamming |
| U.S. Department of Veterans Affairs | I01-BX004031 | Dudley W Lamming |
| National Institute on Aging | AG050135 | Nicole E Richardson<br>Dudley W Lamming |

| | | |
|---|---|---|
| National Institute on Aging | AG051974 | Nicole E Richardson<br>Dudley W Lamming |
| National Institute on Aging | AG056771 | Nicole E Richardson<br>Dudley W Lamming |
| National Institute on Aging | AG062328 | Nicole E Richardson<br>Dudley W Lamming |
| National Institute on Aging | AG000213 | Nicole E Richardson |
| School of Medicine and Public Health, University of Wisconsin-Madison | | Dudley W Lamming |

The funders had no role in study design, data collection and interpretation, or the decision to submit the work for publication.

## Author contributions

Sebastian I Arriola Apelo, Formal analysis, Supervision, Investigation, Writing - review and editing; Amy Lin, Jacqueline A Brinkman, Emma Meyer, Mark Morrison, Jay L Tomasiewicz, Cassidy P Pumper, Emma L Baar, Nicole E Richardson, Mohammed Alotaibi, Investigation; Dudley W Lamming, Conceptualization, Resources, Formal analysis, Supervision, Funding acquisition, Writing - original draft, Project administration, Writing - review and editing

## Author ORCIDs

Dudley W Lamming (iD) https://orcid.org/0000-0002-0079-4467

## Ethics

Animal experimentation: All animal procedures conducted at the William S. Middleton Memorial VA Hospital were approved by the Institutional Animal Care and Use Committee of the William S. Middleton Memorial Veterans Hospital. (Assurance ID: D16-00403).

## Decision letter and Author response

Decision letter https://doi.org/10.7554/eLife.56177.sa1
Author response https://doi.org/10.7554/eLife.56177.sa2

# Additional files

### Supplementary files

• Supplementary file 1. Supplementary Tables S1-S4. Supplementary Table S1: Survival of mice plotted in *Figures 1A* and *2A*.Supplementary Table S2: Survival of mice plotted in *Figures 1B* and *2B*. Supplementary Table S3: Survival of male mice plotted in *Figure 1C*. 22 animals were removed for a cross-sectional analysis and censored ('0'). Supplementary Table S4: Survival of female mice plotted in *Figure 2C*. 23 animals were removed for a cross-sectional analysis and censored ('0'); one additional animal was last recorded at 592 days and censored at that age.

• Transparent reporting form

### Data availability

All data generated or analyzed during this study are included in the manuscript and supporting files.

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
