## [Decision Letter]

**Acceptance summary:**

Dietary and genetic manipulations in mice can produce strong modulation of organismal physiology, and many of the variables involved in behaviors related to feeding. The studies reported in your article expand on your previous work by undertaking a dissection of the role and interplay of hepatic mTORC2 in key aspects of physiology and the regulation of lifespan using a genetic model of hepatic deletion in combination with gonadectomy. These studies add to our understanding of crosstalk between different major signaling systems in the regulation of lifespan.

**Decision letter after peer review:**

Thank you for submitting your article "Ovariectomy uncouples lifespan from metabolic health and reveals a sex-hormone dependent role of hepatic mTORC2 in aging" for consideration by *eLife*. Your article has been reviewed by three peer reviewers, and the evaluation has been overseen by a Reviewing Editor and Jessica Tyler as the Senior Editor. The reviewers have opted to remain anonymous.

The reviewers have discussed the reviews with one another and the Reviewing Editor has drafted this decision to help you prepare a revised submission.

Summary:

Dietary and genetic manipulations in mice can produce strong modulation of organismal physiology, and many of the variables involved in behaviors related to feeding. In this manuscript Dr Lamming's group continues to expand on their previous work and undertake a careful dissection of the role and interplay of hepatic mTORC2 using a genetic model of hepatic deletion in combination with gonadectomy. Apelo et al. examine in more detail the role of hepatic deletion of Rictor on lifespan and healthspan.

The premise of the study was that males, but not females, of this genotype were short-lived, leading to the current project involving the effects of gonadectomy. The authors build on previous reports from the lab showing that deletion of mTORC2 reduces lifespan in male mice, but not female mice. Further, the authors try to tie this to the known difference in lifespan extension with rapamycin with females benefiting more than males. In this study, authors find that castration of males does not affect the male shortened lifespan of mTORC deletion mice, but surprisingly ovarectomy shortens the lifespan of mTORC deletion female mice (but not WT females) in mice that die of non-cancer reasons (but not overall). They also show sex effects of gonadectomy on metabolic function.

While the studies performed are a needed undertake, the quality of the work presented in this manuscript is high, and measuring healthspan across the lifespan in mammalian studies is of high relevance, some major concerns remain:

1) A major complicating factor to this study is that both females and males lacking Rictor in the liver are short-lived.

2) The determination of cancer or no cancer in these mice is unclear. Background strain of C57BL/6 has been reported previously 50-75% mice at death with lymphoma which could be challenging to determine by gross necropsy. Are the "cancer" animals those that present with solid tumors? From a pathological standpoint this is very different than "cancer" vs. "no cancer". I.e., it would be unlikely so many mice of this strain are dying of "no cancer".

3) The beneficial effects of ovariectomy on lifespan in females is surprising based on previous studies (including some from the 1960s) suggesting ovariectomy shortens rodent lifespan. How do authors reconcile these findings based on previous studies?

4) Similarly, previous studies suggest castration improves glucose metabolism (unlike shown here) and ovariectomy impairs (but here no effect). The effects or lack of effect on L-RKO are interesting, but in light of differences in results in WT animals it's hard to determine interpretation.

5) Statistical analysis – authors provide rationale for determining sample sizes, though oddly provide a range (20-30) rather than a specific number per group as a result of power analysis. Regardless, those group sizes decrease when segregating between "cancer" and "non-cancer". Is there then sufficient power to determine, for example, that ovariectomy of L-RKO actually alters death in a subset of "non-cancer"?

6) It is not until the Discussion that the survival differences between females in this study and the prior one are explained. This makes it hard for the reader. It should be addressed much earlier in the manuscript for clarity.

7) The authors equate healthspan to metabolic parameters, which seems to be a narrow interpretation. They should either include other component measures of healthspan or switch to metabolic health (or some other term that more closely describes what is being measured). With the limited number of metrics analyzed, it is hard to substantiate the notion that ovariectomy improves survival but negatively impacts healthspan.

8) It would be important to show the degree of deletion that is achieved by this genetic cross, does it produce the same level of change in males and females? If not available, did the authors measured any activity changes down from MTORC2? How does the female control lifespan compare to other studies published on B6 females? Any changes in litter weights, sizes or ratios males/females in the genetic crosses?

9) It would be important to report, if you have it, food consumption. Are some of the differences in body weight linked in any way to food intake?

10) Besides cancer, any other predominant disease that the animals are dying off/with?

11) For all the reports on metabolic parameters it would be great if the authors could set the Y axes to the same max values, it would make easier to compare the age and sex related changes in Figure 3A-D and Figure 4A and B, E and F.

12) It is not clear how to do this but the manuscript is difficult to read through because of the different cohorts of mice and sexes. The authors should consider ways to improve this. One possibility would be to discuss males and females separately in the Results and then bring the differences out in the Discussion. Not sure if this would improve clarity, but it might be worth a try.

---

## [Author Response]

[…] While the studies performed are a needed undertake, the quality of the work presented in this manuscript is high, and measuring healthspan across the lifespan in mammalian studies is of high relevance, some major concerns remain:1) A major complicating factor to this study is that both females and males lacking Rictor in the liver are short-lived.

We apologize for the confusion, but actually, this is not quite correct from a statistical standpoint – there is only an effect of liver Rictor loss on the overall survival of male mice. As we state, “when we looked at the total population of female mice in our study, the negative effect of *Rictor* loss on Sham-treated female mice was masked, and there was no overall negative effect of *Rictor* loss on survival” (Cox regression analysis: HR Rictor knockout = 1.02, p = 0.92), Figure 2C).

Loss of liver Rictor does impact the survival of a subset of female mice (Figure 2B), which is a major finding of our manuscript. However, while this decrease in midlife survival leads to a trend towards reduced survival of all L-RKO Sham females between 400-800 days in Figure 2C, this does not reach statistical significance as the curves converge again after 800 days. To emphasize this, we have edited the text to state more clearly that “There was also no difference between the overall survival of WT Sham and WT L-RKO female mice (p = 0.153, Wilcoxon rank sum)”.

2) The determination of cancer or no cancer in these mice is unclear. Background strain of C57BL/6 has been reported previously 50-75% mice at death with lymphoma which could be challenging to determine by gross necropsy. Are the "cancer" animals those that present with solid tumors? From a pathological standpoint this is very different than "cancer" vs. "no cancer". I.e., it would be unlikely so many mice of this strain are dying of "no cancer".

We thanks the reviewers for highlighting this issue, which we agree is quite important – in fact we label our figure panels “deaths with cancer” and “cancer not observed at necropsy” rather than “deaths from cancer” or “deaths with no cancer” for this very reason. We have edited the figure legends to make this distinction as well; and we have edited our methodology to better describe our gross necropsy.

Approximately half of the mice had observable cancer during our gross necropsy, which is consistent (but on the lower end) of the range of cancer rates reported for C57BL/6J mice in the literature. We have now edited the Results and Discussion to more clearly note that there could be unobserved cancers as well. We now highlight that an effect of ovariectomy on unobserved neoplasma could contribute to the protective effect of OVX on L-RKO female lifespan, and we now state that “Future carefully designed studies to assess the organismal health of L-RKO mice, with detailed pathology between 400 and 800 days of age, will be needed to fully address these issues.”

3) The beneficial effects of ovariectomy on lifespan in females is surprising based on previous studies (including some from the 1960s) suggesting ovariectomy shortens rodent lifespan. How do authors reconcile these findings based on previous studies?

This is an interesting question – we have revised and expanded our Discussion section to address this. In brief, at least some older studies in rats suffered from the lack of sham surgery control group; a 1967 study suffering from this flaw also observed no difference between the lifespan of OVX mice and OVX mice dosed with estrogen, suggesting the difference in lifespan of wild-type and OVX mice could have resulted from the stress of surgery rather than removal of estrogens. A recent 2018 rat study which we now site reports OVX extends the lifespan of mice.

A 2015 mouse study showed that ovariectomy at 5 months of age shortens lifespan, in contrast, we performed ovariectomy at 3 weeks of age, prior to puberty, and studies in *C. elegans* and grasshoppers support the idea that early life removal of germ cells/ovaries extends lifespan. Thus, timing of ovariectomy may be important as well. We thank the reviewers for urging us to address these important points.

4) Similarly, previous studies suggest castration improves glucose metabolism (unlike shown here) and ovariectomy impairs (but here no effect). The effects or lack of effect on L-RKO are interesting, but in light of differences in results in WT animals it's hard to determine interpretation.

We thank the reviewer for this question, but it is based on an incorrect premise and a misunderstanding of our results. We have expanded our discussion of glucose homeostasis substantially to ensure the reviewer and reader can better understand our results, citing additional recent findings.

First – we do observe impaired glucose metabolism in OVX mice as expected and now state in the Discussion “these overall positive effects of ovariectomy on longevity do not promote healthspan, with ovariectomized mice suffering from adiposity and disrupted blood glucose homeostasis.” You can see this specifically in panels Figure 5C and D – the blue curve for WT OVX mice is significantly higher than the curve for WT Sham mice, and the AUC is significantly higher. There is similarly an effect of OVX on fasting insulin and calculated HOMA2-IR (Figure 5F).

Second, we see no effect of castration on glucose tolerance or insulin sensitivity, but we did not expect to see a positive effect – if anything, we expected an impairment as we expected body composition to worsen. Our results, showing no effect of castration on glucose homeostasis in young or aged mice, are in agreement with recent reports showing no effect of castration on glucose homeostasis in mice eating a control diet (Garratt et al., 2017; Harada et al., 2016). This latter paper reports an impairment of glucose homeostasis in castrated mice (unlike what the reviewer mentions) upon high-fat diet, but we did not examine high fed diet fed mice here.

5) Statistical analysis – authors provide rationale for determining sample sizes, though oddly provide a range (20-30) rather than a specific number per group as a result of power analysis. Regardless, those group sizes decrease when segregating between "cancer" and "non-cancer". Is there then sufficient power to determine, for example, that ovariectomy of L-RKO actually alters death in a subset of "non-cancer"?

As now better discussed in the transparent reporting guidelines we targeted a group size of 20-30 mice per group in order to provide 84-95% power to detect a 15% change in lifespan. We selected a target range as mouse breeding of genetically engineered mice is never precisely mendelian and we then needed to perform surgery on the mice. We achieved this target range for all groups.

To address the second concern we performed a power calculation using the actual mean and standard deviations observed here for the L-RKO Sham and L-RKO OVX lifespans. We find that we had 85% power to detect the effect size we observed (α = 0.05).

6) It is not until the Discussion that the survival differences between females in this study and the prior one are explained. This makes it hard for the reader. It should be addressed much earlier in the manuscript for clarity.

We thank the reviewers for this comment, and we have edited the manuscript to highlight this result in the Introduction and Results section.

7) The authors equate healthspan to metabolic parameters, which seems to be a narrow interpretation. They should either include other component measures of healthspan or switch to metabolic health (or some other term that more closely describes what is being measured). With the limited number of metrics analyzed, it is hard to substantiate the notion that ovariectomy improves survival but negatively impacts healthspan.

We completely agree that this is a limitation, and as suggested by the reviewers we now use the term “metabolic health” in the Abstract and Discussion, and state that “A limitation of this study is that we did not perform a comprehensive assessment of healthspan, focusing on metabolic parameters.”

8) It would be important to show the degree of deletion that is achieved by this genetic cross, does it produce the same level of change in males and females? If not available, did the authors measured any activity changes down from MTORC2? How does the female control lifespan compare to other studies published on B6 females? Any changes in litter weights, sizes or ratios males/females in the genetic crosses?

We thank the reviewer for this important question. As we have shown previously, and we now include in the Introduction, the genetic cross achieves equivalent deletion of liver mTORC2 in both males and females, reducing overall expression of liver RICTOR protein by approximately 80% through the essentially complete deletion of *Rictor* in hepatocytes and reducing phosphorylation of mTORC2 substrates by a similar percentage (Lamming et al., 2014; and Lamming et al., 2014).

The control lifespan of our female WT Sham mice are equivalent to C57BL/6J female mice from The Jackson Laboratory (Our female WT Sham median lifespan: 853; Jackson: 866) and are consistent with the lifespans of two other B6 mouse lifespan studies not involving surgery recently published by our lab (Yu et al., 2019; Chellappa et al., 2019).

9) It would be important to report, if you have it, food consumption. Are some of the differences in body weight linked in any way to food intake?

We thank the reviewer for this important question. We have included a new figure, showing that ovariectomy leads to alterations in energy balance and fuel source utilization. At least in 12 month old animals, ovariectomy seems to promote obesity as a combination of reduced physical activity and reduced energy expenditure, and promote utilization on lipids as an energy source.

10) Besides cancer, any other predominant disease that the animals are dying off/with?

While a spectrum of other age-related diseases is undoubtedly present in C57BL/6J mice, there was no other predominant cause of death that we were able to identify during our gross necropsies.

11) For all the reports on metabolic parameters it would be great if the authors could set the Y axes to the same max values, it would make easier to compare the age and sex related changes in Figure 3A-D and Figure 4A and B, E and F.

This was a great suggestion, and we have now made all Y axes that represent the same measurements have the same range and min/max values in Figures 3, 4, and 5.

12) It is not clear how to do this but the manuscript is difficult to read through because of the different cohorts of mice and sexes. The authors should consider ways to improve this. One possibility would be to discuss males and females separately in the Results and then bring the differences out in the Discussion. Not sure if this would improve clarity, but it might be worth a try.

We thank the reviewers for this suggestion – we actually started writing the manuscript this way and also tried a few other constructions, but ultimately settled on the present order as the clearest. We hope the other clarifications and edits we have made have helped to improve flow and readability of our description of this admittedly complex study.